# A Qualitative Exploration of a Biopsychosocial Profile for Experiencing Sexual Harassment and Abuse in Sports

**Mercede van Voorthuizen [1], Irene Renate Faber [2,3], Daphne van de Bongardt [1] and Nicolette Schipper-van Veldhoven [2,4,5,6,*]**

1   Erasmus School of Social and Behavioural Sciences, Erasmus University Rotterdam, 3062 PA Rotterdam, The Netherlands; vanvoorthuizen@ese.eur.nl (M.v.V.); vandebongardt@essb.eur.nl (D.v.d.B.)
2   Research Centre Human Movement and Education, Windesheim University of Applied Sciences, 8017 CA Zwolle, The Netherlands; i.r.faber@windesheim.nl
3   Institute of Sports Science, University of Oldenburg, 26129 Oldenburg, Germany
4   Faculty of Behavioural, Management and Social Sciences, Mathematics and Computer Science, University of Twente, 7522 NB Enschede, The Netherlands
5   Faculty of Electrical Engineering, Mathematics and Computer Science, University of Twente, 7522 NB Enschede, The Netherlands
6   Netherlands Olympic Committee and Netherlands Sports Confederation (NOC*NSF), 6816 VD Arnhem, The Netherlands
*   Correspondence: n.schippervanveldhoven@windesheim.nl

**Abstract:** The purpose of this study was to explore a biopsychosocial profile for experiencing sexual harassment and abuse in sports. A qualitative approach was used; data were collected from semi-structured in-depth interviews covering seven cases of sexual harassment and abuse in sports in the Netherlands. The interview transcripts were analysed and aligned with the biopsychosocial model. The results reveal biological (i.e., aged under 18, sex, and sexual orientation), psychological (i.e., high degree of naivety, altruism and agreeableness, low self-esteem, perfectionism, emotional or disorders) and social factors (i.e., poor or negative relationship with parents, social pressure to perform, incest at home, social isolation, elite sports and too much power of a single trainer/coach) that can contribute to the risk of experiencing sexual harassment and abuse in sports. These findings provide important directions for prevention and recognition in sports practice and future research.

**Keywords:** sexual harassment; sexual abuse; biopsychosocial model; organised sports; qualitative study

## 1. Introduction

The focus on sexual harassment and abuse (SHA) in sports dates back to the 1990s. At that time, a number of cases in the media made it clear that SHA unfortunately also happened in the sports context (Brackenridge 1997; Brackenridge and Kirby 1997; Volkwein et al. 1997). The attention paid to SHA in sports increased from that time. In the present study, sexual harassment is defined as any form of sexual behaviour or sexual approach, verbal, non-verbal or physical, intentional or unintentional, which is experienced as unwanted or coerced by the person undergoing it (Vertommen and Schipper-van Veldhoven 2012; Vertommen et al. 2016a). Sexual abuse involves intentional physical sexual behaviour that is experienced as coerced or unwanted, where consent is not or cannot be given (e.g., when under the influence of alcohol or drugs; International Olympic Committee 2007; Johansson 2013). It became clear that SHA in sports could lead to serious consequences such as self-image distortion, anxiety, obsessions, phobias, sexual dysfunction, substance abuse, eating disorders, dissociative disorders, post-traumatic disorder, depression, self-harm, borderline personality disorder, psychosis, and suicide (Briere and Elliott 2003; Maniglio 2009; Molnar et al. 2001). Consequently, policy makers tried to prevent future cases by different strategies.

The Netherlands had been a pioneer in terms of (prevention) policies on SHA since 1996 (Mergaert et al. 2016; Moget et al. 2012; Schipper-van Veldhoven et al. 2015). In that year, three Dutch (ex) judokas filed a complaint against their coach. In response, the Netherlands Olympic Committee*Netherlands Sport Confederation (NOC*NSF) introduced prevention strategies and instruments such as preparing a 'code of conduct' in 1997, appointing confidential counsellors in 2005, implementing unambiguous disciplinary regulations in 2008, making an official 'statement about behaviour' mandatory for trainers and coaches in 2011 and the development of a toolkit for realizing a safe sports environment (Schipper-van Veldhoven et al. 2015).

However, in 2015, a large prevalence study showed that approximately 1 in 10 athletes still dealt with SHA in sports before their 18th birthday in the Netherlands (Vertommen et al. 2016b). Moreover, the instruments mentioned do not appear to be implemented in many of the local sport clubs in which most Dutch youngsters practice their sport (Schipper-van Veldhoven 2017; Schipper-van Veldhoven et al. 2015; Serkei et al. 2012). In addition, there is a lack of evidence supporting the impact of the different strategies/instruments. These shortcomings emphasize the need for a more effective and accessible prevention policy for SHA in sport (Romijn et al. 2016; Schipper-van Veldhoven 2017).

Accordingly, new initiatives were undertaken. First, the NOC*NSF appointed an independent committee to investigate the issues regarding the policies and procedures. In 2017, the Committee concluded that there was a need for a more active approach to incidents and prevention (De Vries et al. 2017). A so-called 'duty to report' was recommended. Second, the European project VOICE (Hartill et al. 2019) was launched to better arm the sports world against SHA and to strengthen the integrity of sport by understanding the conditions and consequences of SHA in sports through the perspectives of those affected. The focus of this project was solely on the victim/survivor's story. Semi-structured interviews were conducted with victims/survivors of SHA in sport and from these stories the authors reflected on the causes, nature, and circumstances of SHA in the sport context, before making recommendations for prevention. Third, several support/solidarity networks developed over time. An example is the Dutch Centre for Sport and Safety (https://centrumveiligesport.nl/, accessed on 8 June 2022) that recently set up a comprehensive national campaign called "To be silent or to talk?" (Zwijgen of praten?). There were also initiatives from survivors who want to help others (e.g., https://destilteverbroken.nl/, accessed on 8 June 2022).

Nevertheless, it must be concluded that even though previous (research) projects and the aforementioned new initiatives have been undertaken, so far, hardly any project or research has focused on risk factors for experiencing SHA in sports. Yet, it is precisely the insight into these factors that could contribute to improved prevention strategies (Cense and Brackenridge 2001). So far, the small number of studies on SHA in sports have focused mainly on the coach–athlete relationship (Bjørnseth and Szabo 2018; Johansson et al. 2016; Nielsen 2001; Stirling and Kerr 2009) with 'grooming' as a theoretical explanatory model; the process by which a perpetrator slowly gains a potential victim's trust before systematically breaking down interpersonal barriers (Brackenridge and Fasting 2005). Research into the origin of SHA in sports was thus mainly understood from a social perspective (Moget and Weber 2008). However, it also seems sensible to conduct research from an individual perspective, to better understand how SHA arises. Why are some young people more prone to experiencing SHA than others? As researchers, we also want to strengthen the athletes themselves in how to act when there is a suspicion of a SHA process. Although the prevention strategy of SHA in the Netherlands was based on the temporal model on risk factors for SHA (Cense 1997) in which the researchers identified three clusters of risk factors associated with (1) the coach, (2) the sport context, and (3) the athlete (Cense 1997), most of the actions since then have been focused on policy and the entourage, not on the empowerment of athletes themselves (other than listening to their stories) (Schipper-van Veldhoven et al. 2015). Cense's model was based on the idea that several sequential phases precede the sexual abuse of children and young adults: 1. motivation; 2. overcoming

internal inhibitions; 3. overcoming external inhibitions; and 4. overcoming the athlete's resistance. In the last step of the sequential process (step 4) the abuser will try to undermine the athlete's resilience. Indicators are actions or events which cause the child to feel insecure, helpless or abandoned. A weak social position and low self-esteem are examples of indicators that increase the risk of abuse (Moget et al. 2012).

The present study, therefore, focuses on the athlete and the potential risk factors for experiencing SHA in sports by using a biopsychosocial perspective. Due to the convergence of individuals, families, and their social environment within the sports context, it is likely that the development of sexual harassment and abuse in sports arises from an interaction of various factors. The biopsychosocial model (see Figure 1) based on Engel's (1977) model (e.g., Whitbourne and Whitbourne 2010; or more generally: the multisystem perspective) is a well-known and frequently used theoretical model in the discipline of pedagogical sciences (focusing on child development, upbringing and education) and also in the research fields of both general (physical and mental) health (e.g., Lehman et al. 2017; Lindau et al. 2003) and sexual health (e.g., Laan et al. 2021), both of which are closely linked to the topic under investigation in the present study. The general idea in this theoretical model is that an individual's development (in youth or throughout the course of their life) is shaped by complex interactions between biological factors (such as age and sex), psychological factors (such as personality and self-esteem) and social factors (such as family context and other social relationships; Adler 2009; Borrell-Carrió et al. 2004; Whitbourne and Whitbourne 2010). This line of thought is also very relevant to the general field of sports psychology and pedagogy, and the specific topic under investigation in the present study; as we argue, experiences with SHA in the context of sports can also best be understood through various relevant biological, psychological, and social factors. It is well known that the sports context is characterized by a number of traditional values that can give rise to the emergence and persistence of SHA: unequal power relations between authority figures (such as coaches) and athletes, pressure to perform, often requiring physical contact and participation at a young age (Cense and Brackenridge 2001; Bjørnseth and Szabo 2018; Vertommen et al. 2016a; Vertommen and Schipper-van Veldhoven 2012; Volkwein et al. 1997). In combination with these factors within the sports context, individual factors, as well as social factors outside sports, could play an important role in experiencing SHA in sports.

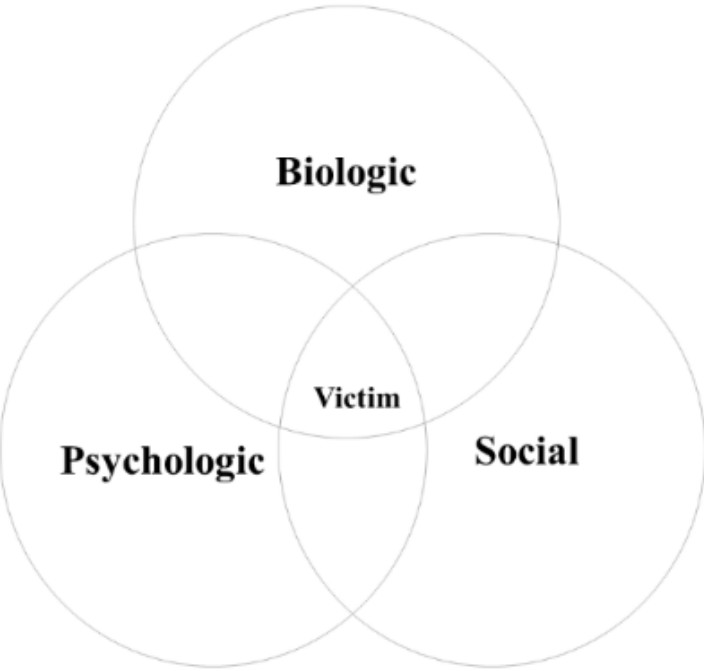

**Figure 1.** Biopsychosocial model.

The aim of the present study was to gain insight into potential biopsychosocial risk factors for experiencing SHA in sports to be able to make prevention recommendations to the young people themselves, as well as their parents, trainers, and administrators, and create associations for recognising the vulnerability of young people for SHA in sports and to protect them from becoming a victim of a (potential) perpetrator. The relevance of using a biopsychosocial perspective is also shown in previous studies which applied this model in their examinations of the (long-term) impact of (early-life) experiences with both general and sexual intimidation, violence, or abuse, as well as strategies aimed at prevention or harm reduction (e.g., Coates 2010; Edwards et al. 2014; Murphy et al. 2014; Pereira-da Silva et al. 2017; Rahill et al. 2020). Based on findings that suggest the social, psychological and biological consequences of abuse interact in complex ways over the lifespan of victims of child abuse, advice has been given to strengthen potential victims and to teach them: (1) arousal reduction strategies (psychological), (2) strategies that strengthen the corpus callosum (biological), and (3) how to develop social skills and new relationships (social) (Whitbourne and Whitbourne 2010).

It must be emphasized that by conducting this study it is by no means intended to place the responsibility of experiencing SHA on the victim/survivor. It is intended to get to grips with factors influencing athletes' vulnerability (risk factors in the SHA process). We chose to use 'victim/survivor' to do justice to the various lived experiences of the individuals in our sample and the population they represent.

## 2. Materials and Methods

### 2.1. Design

For this explorative qualitative study, semi-structured in-depth interviews were conducted with victims/survivors of SHA in the context of sport. This approach was selected to unravel the potential biopsychosocial factors for experiencing SHA based on the premise that the victim/survivor's own story would contain the most valuable information to gain insight into the origin of SHA in sports, allowing the exploration of the subject (Boeije 2014). The study was conducted in accordance with the Declaration of Helsinki. As part of the regular ethical procedures, the study design and data collection procedures were evaluated against the Ethical Decision Tree and approved (Local Ethics Committee of Windesheim University of Applied Science (Zwolle, The Netherlands)).

### 2.2. Participants

Seven victims/survivors of SHA in sports (five women and two men, aged between 22 to 57) participated in this study (Table 1). The participants' age at which SHA occurred ranged from 6 to 56 years. At the time of their SHA, two participants practiced a team sport. The other participants were active in an individual sport, of which two were at the elite level (i.e., Dutch national selection/team and competing at the highest national or international level).

**Table 1.** Demographics of the participants.

| Participant | Age Now (Time Study) | Age Start SHA | Sex | Individual or Team Sport | Recreational or Elite Level |
|---|---|---|---|---|---|
| 1 | 57 | 14/15 | male | team | recreational |
| 2 | 28 | 16 | female | individual | elite |
| 3 | not known | 10 | female | individual | elite |
| 4 | 22 | 17/18 | female | individual | recreational |
| 5 | 17 | 6 | female | individual | recreational |
| 6 | 57 | 56 | female | individual | recreational |
| 7 | not known | 16 | male | team | recreational |

*2.3. Procedure and Data Collection*

Participants were recruited by a call for participation placed in (local) newspapers and on (sports) websites. One member of the research group (MvV) checked their eligibility for the study (age of 18 years or older and victim/survivor of SHA in sports). All participants gave their informed consent before they participated in the study. Participants did not receive any reward for their participation.

The semi-structured in-depth interviews were conducted based on a topic list covering a victim/survivor's background information, family history and childhood, sport history, personality, the harassment/abuse and a reflection on what happened. In one case, the parent (as a spokesperson for her daughter) of a victim/survivor was interviewed instead of the victim/survivor herself due to her current personal circumstances and vulnerability due to emotional disabilities. One of the victims/survivors did not want to participate in a real-life interview, but provided a written elaboration instead. All real-life interviews were led by a confidential counsellor from NOC*NSF and one of the authors (MvV) who underwent comprehensive interview training prior to the study. The average duration of the interviews was two hours. The interviews were recorded with an audio recorder. The digital audio recordings of the interviews were transcribed verbatim by one of the authors (MvV).

*2.4. Analyses and Coding*

The coding and content analysis of the relevant parts of the interview transcripts was conducted with qualitative data analysis software ATLAS.ti (Scientific Software Development GmbH; version 9.0.24 for Windows, 2021) and performed according to the principles of the grounded theory approach (Corbin and Strauss 1990), due to the combination of the theory-driven and data-driven approach, and because the processes of coding and analysis were iteratively intertwined.

Firstly, fragments related to the research question were selected from the interview transcripts. This selection was ordered by concept, and fragments belonging to the same category were grouped (Boeije 2014). Within the subject, each fragment was given a code and a summary name (open coding), which were drawn up based on the theory. If these theoretical codes did not sufficiently cover the load of the transcripts, additional codes were formulated. These codes were then integrated into central categories (axial coding). When the central categories were clear, connections were made (selective coding). After coding of the first interview transcript, consultation took place with the co-researchers in a debriefing session to check the coding for correctness and accuracy. Subsequently, all analyses were supervised by the co-researchers, to ensure the reliability of the coding.

**3. Results**

*3.1. Description of Experiences*

The participants described the following experiences with SHA in sports: unwanted physical contact, bullying based on gender, sexually explicit videos, groping, physical violence, unwanted exposure to genitals or pornography, unwanted photography or filming, unwanted kissing or other sexual acts, and rape. The duration varied from one to two years. The perpetrators described by all seven participants were male. The exact ages of the perpetrators at that time are unknown. In five cases the perpetrator was the participant's coach/trainer who was clearly much older than the participants. In one case, the perpetrator was a fellow athlete within the sports club, whose age was similar to the participant. In one case, the participant had dealt with both situations (coach/trainer and fellow athlete).

The participants described how they reacted immediately after the incident of SHA. Most of the participants' initial reaction was to resist the perpetrator. However, according to the participants, resisting turned out to be counterproductive. In most cases, the participant gave up and let it happen to let it pass quickly. Participant 4 described: *"Well in the end I just shut down. Yes, in the beginning I said no don't do it and I was really fighting against it and then I*

*couldn't do it anymore. Then I didn't know what to do anymore. And then I thought, apparently this is how it should be. And I just have to accept that.*" One participant managed to prevent the perpetrator from continuing his advances (i.e., putting his hand on the victim/survivor's leg) by shouting loudly. For example, participant 7 said: "*I jumped up angrily and said in a loud voice: 'You have 10 s to get out of this house or I'll knock you out' I was furious. Coach went downstairs and to the front door like a rocket. I still yelled after him: 'If I ever see you here or anywhere again, I will kill you!' Coach literally jumped on his bike, almost lost his balance and swung hard down the street.*"

Although the first responses revealed many similarities, the 'long-term' perceptions of the experiences seem to differ between the participants. For example, there was one participant who did not experience most of the experiences as unpleasant. He described the experiences as exciting in a positive sense and never felt unhappy about it. This participant said, when looking back in time, "*I find it very annoying that I have been initiated into sexual acts by the wrong person.*" This participant was also forced by the perpetrator to perform sexual acts harassing another boy "*within that I also know that that one time I hurt that boy in my experience, because he cried*". This participant also realized only later that anal sex at such a young age is not appropriate and that his search for his identity made him vulnerable. The other participants declared that they thought that their SHA experiences apparently were 'part of the deal'; i.e., these experiences were considered as required and/or integral components of sport performance at the elite level. At that time, they felt like they needed to accept this kind of behaviour. Some illustrative quotes are: "*And then I [participant 4] thought apparently that this is how it should be. And I just have to accept that.*", "*Somehow, I [participant 3] thought it was part of it. That that kind of things happen between trainers and pupils.*", and "*I [participant 2] just thought it was normal. Yeah, I thought it was just normal. I thought it was the way things worked [in sports].*" There were also two participants who said that afterwards they were unsure about whether the SHA had actually happened or whether they had made it up themselves. "*You file a report, but in the meantime I was really unsure of what happened really happened? Or was it not my experience? That you just start to doubt yourself.*"

### 3.2. Biopsychosocial Factors

A number of factors that may have contributed to experiencing SHA in sports were distilled from the data. These factors are presented below while using the biopsychosocial model as framework.

### 3.2.1. Biological Factors

For six of the seven participants their experience of SHA in sports occurred before the age of 18. One of the participants was sexually harassed at the age of 56. Participant 2 said she was sexually harassed precisely because of her sex (i.e., being a girl/woman): "*So then I was the only girl as a 16-year-old between how old those boys were I think between 18 and 26. . . . Only it turned out that those boys were not so nice together.*" Participant 1 suggested that his search for his sexual orientation probably played a role in the development of SHA: "*I think that gave more impetus to my search for who I was or am*", "*And I think that because of that search I was very vulnerable.*" and "*In retrospect, I think I went along with it, because at that age I also started to notice that I was different from others. That I had the idea that I liked boys more than girls. And yes, you will get male attention. Although from a man of the wrong age.*"

### 3.2.2. Psychological Factors

From the participant's stories, there were signals of a relatively high degree of naivety, altruism and agreeableness. The participants described their personality in the period prior to and/or during the SHA as naive, connecting, social, everyone's friend, being ready for others at the expense of themselves, trusting, respectful and caring. Participant 3 explicitly stated in her description that, in retrospect, her helpful behaviour seems to have played a role in her experience of SHA: "*Well, that I really ignored myself for others. If other people had to*

*cry or were sad, yes, then I thought that was terrible . . . And um that made it possible to happen. I thought he was really alone in the world, and I had to solve that.*" Participant 2 described herself as quiet and naive: "*And I was then not that articulate either. I should have just bitten off a lot more. . . . You just let it all come over you.*" and "*I was naive. . . . I still find myself naive at some times.*"

Moreover, the stories tell of the participants generally having low self-esteem in the period before SHA. For example, participant 4 said: "*That it was all not good enough and that I should have done better. . . . Then I was very insecure. I was very quiet. Did not say much.*" Participant 3 described a combination of perfectionism and insecurity: "*I am just a terrible perfectionist and I always wanted to do it really well. So, I have always been an insecure person.*" These factors (and their interaction) seem to have resulted in the situation that unpleasant experiences were not noticed or addressed by the participants. To illustrate, participant 2 reported: "*No, I'm not sharp. And especially not then. Now these days I stick up for myself. But not then.*"

The mother of one of the victims pointed out her daughter's emotional disabilities made her easy to influence and manipulate. The participant was not able to sense whether behaviour is permissible or not*: "But with my daughter [participant 5], it just isn't like that. She knows no boundaries. She cannot sense these things at all.*" According to the mother, this presumably made the participant easy prey for SHA in sports. Participant 6 revealed that due to the incestuous situation at home, she developed a dissociative identity disorder. According to the participant, this disorder influenced the participant's behaviour and perception of SHA in the sport context: "*He rubbed my thigh with his hand and was just not yet in my crotch. I myself fell into dissociation. That means I see it happen, want to scream loudly, but it won't work. I turn everything off.*"

### 3.2.3. Social Factors

The family context of the participants varied. Descriptions ranged from warm and stable, to poor contact or negative relationships with parents, pressure to perform and incest at home. Two participants described a stable family context. Participant 2 reported: "*I have a very good time at home. A good relationship with both of my parents. They are still together.*" Participant 1 emphasized his stable home situation several times: "*I never lacked attention and love from my parents at home.*" Both participants did not consider their family contexts to be a risk factor for SHA in the sports context. In contrast, other participants believed that their home situation contributed to their experience of SHA. Participant 3 said that her parents were seldom at home: "*Yes, my parents worked a lot. . . . So, we were often at day-care, and we had a lot of babysitters at home.*" Participant 4 described problems within the family that may have played a role in the development of her experience of SHA: "*I had quite a few problems at home at the time. It didn't go very smoothly, and I was able to tell my story to him.*", "*Well with my mother, things didn't go well with work and such. I don't know exactly what she had, but you could think in a depression-like thing, which put a lot on my shoulders.*" and "*My father had a wife at the time. . . . He didn't like us and was always negative towards us. So, I couldn't go to my father at the time either.*" Both participant 3 and 4 mentioned these circumstances as reasons to seek someone's attention. Participant 4 described being pressured to perform from an early age. This pressure to perform was perceived as negative by the participant. It emerged from the participant's description that the pressure to perform from home was a reason for her to seek support within the sports association. The participant described: "*Well I know from childhood there was pressure on my sister and me to perform. My sister who is just much smarter than I am. She studies at the university and she really has more cognitive 'things' than I have. And my mother also went to university. She also worked at a university. So she also had the intelligence and she also thought that performing was very important. . . . And that's why I looked for a lot of support outside of home. And that's what I got out of that boy who eventually sexually abused me.*" According to participant 6, the incestuous situation at home had a huge impact on the participant's life, which led to dissociative behaviour as mentioned above.

The stories of the participants also showed differences regarding friends. Participant 1 and 6 indicated that they had no friends and felt lonely at the time. They described themselves as socially isolated. Participant 1 said "*I think that was my general problem. Few friends both at school and at the sports club.*" According to these participants, the lack of friends made them more susceptible to the attention of their coach. The other participants did not report a lack of friends or even spoke about having a group of friends. Participant 2 said "*I dare say that about myself. I used to be popular in school. I always had friends. I was never alone.*" In addition, participant 3 said "*Every day I had someone to meet up with*" and participant 4 "*Really nice friends that I got along well with.*"

The sports context was also mentioned in several ways as a potential social risk factor for experiencing SHA. The participants' stories showed that SHA takes place in sport at both a recreational and elite level. However, the SHA experiences of the two participants in elite sport lasted longer and seemed to have had a greater impact. The perfectionism and passion of elite athletes and their surrounding is often so great that everything has to give way to make great performances possible. For example, participant 3 said: "*I was sure. [Sports] was everything to me. I was going to give everything, but really everything for that.*" Moreover, elite athletes are often on tour with their coach, creating opportunities for harassment and/or abuse. Participant 3 explained: "*He was looking for a way to be alone with me. He made up trips. He really wanted to go with me and did his very best to accomplish this.*" Additionally, the mother of participant 5 said: "*You weren't allowed to come into the locker room. You weren't even allowed to be in the [sports facility] itself, because that would distract the kids too much.*" For participant 2, the perpetrator had a lot of power, since he could determine her sports career: "*Because that man determines everything. He determines whether you will receive money from the sports institute. Whether you can use the facilities. He sends you to competitions. Everything. He decides whether you go to the Olympic Games or not. Yes, he determines your career.*"

### 3.3. Biopsychosocial Profile

The analysis of all of the participants' stories revealed factors of the different levels within the biopsychosocial model that seem to have played a role in the origins of their experiences of SHA in sports. Figure 2 summarizes the biopsychosocial factors distilled from the transcripts using the biopsychosocial model.

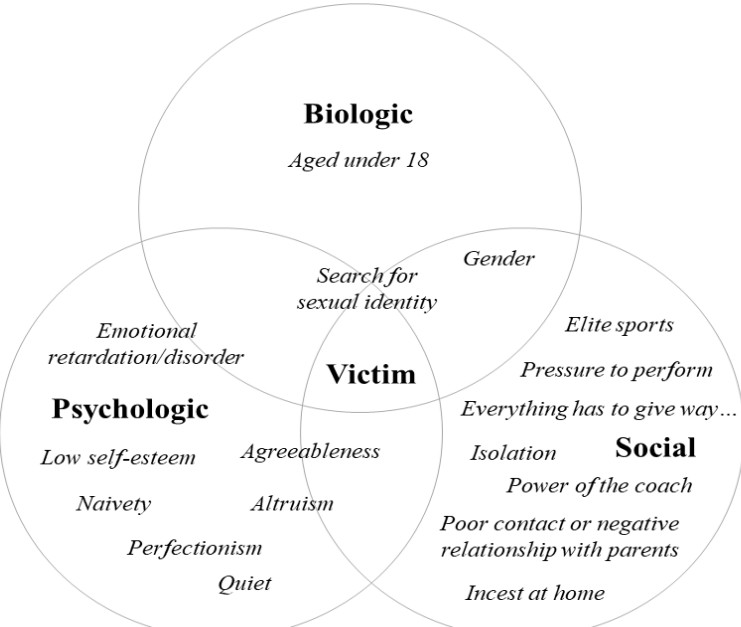

**Figure 2.** Schematic overview of possible biopsychosocial risk factors for experiencing SHA in sports, as determined from the current study.

It is important to acknowledge that the seven interviewees all showed a unique profile and put different emphasis on each of the factors that they believed have played a role in their own experiences of SHA. Figure 3 shows how profiles can be established for individual cases. For this purpose, the storylines of participants 1, 2 and 3 are used as illustrations.

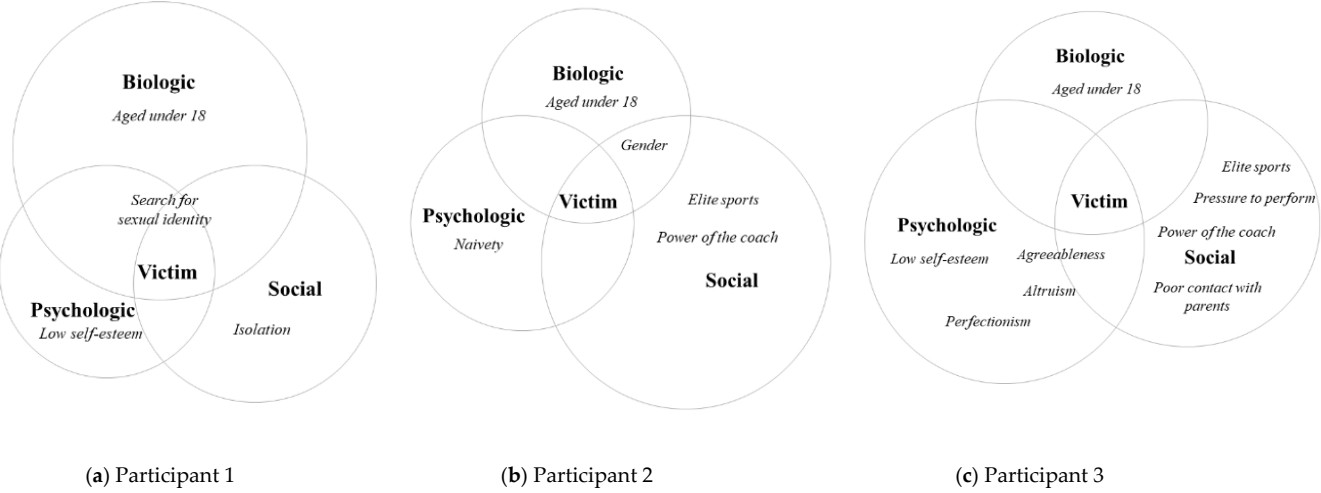

(**a**) Participant 1       (**b**) Participant 2       (**c**) Participant 3

**Figure 3.** Schematic overview of three biopsychosocial profiles.

Participant 1 (Figure 3a) was 14 years old when his experience with SHA in sports started. At that time, he began to doubt his own sexual orientation. He had only a few friends, both at school and at his sports club. His socially isolated position made him vulnerable to attention from his coach. He pointed out that this played a major role in the development of his victimhood.

Participant 2 (Figure 3b) was 16 years old when her experience with SHA in sports started. She considered herself naive at the time, and in retrospect, she said that she should have stood up more for herself. Within the national selection she was the only girl, and was dependent on her coach for a top sport career. This made her, in her own words, vulnerable.

Participant 3 was 11 years old when her experience with SHA in sports started. It soon became apparent that she had talent and could participate at an elite sports level. Her parents worked hard, so her coach took care of everything around sports. She always put herself in second place, with the well-being of others taking precedence over her own well-being. In retrospect, she said that this individual characteristic was probably what contributed to the SHA that happened to her.

This shows that the understanding of SHA in a sports context requires a deep, multi-dimensional analysis of the complex interplay of biopsychosocial risk factors.

## 4. Discussion

The aim of the present study was to explore the biopsychosocial factors that may increase the risk of experiencing SHA in sports, without any intention of victim-blaming. The findings confirm and broaden previous research on potential risk factors for SHA in sports. In the qualitative analysis, the victims/survivors described factors on a biological (age (being a minor), sex, and sexual orientation), psychological (high degree of naivety, altruism and agreeableness, low self-esteem, perfectionism, and emotional disabilities) and social level (poor or negative relationship with parents, social pressure to perform, incest at home, social isolation, elite sports, and too much power of a single trainer/coach) that put them at risk of experiencing SHA in sports. These factors seem to make athletes more vulnerable; perpetrators might sense these characteristics, which makes these athletes a target. This finding confirmed that the onset of SHA in sports is more complex than previously thought. Not only factors within sports, but also multiple characteristics of the individuals and their social environment, seem to play a role.

The biological factors age, sex and sexual orientation, which are defined in the present study, have been related to the development of SHA experiences in previous studies of SHA in sports. These studies found that: (a) both male and female athletes become victims/survivors; (b) the most vulnerable period for experiencing SHA is before the age of 18; and (c) that LGB athletes report a higher risk of being a victim/survivor of SHA in sports (Parent and Bannon 2012; Vertommen et al. 2016b). The current findings complement these findings.

The contribution of the present study to previous research is the attention to psychological factors that may signal potential risk for experiencing SHA in sports. In contrast to low self-esteem (Brackenridge 2001; Butler 2013), personality has not previously been associated with the development of experiencing SHA. Personality traits distinguish underlying individual dispositions that can influence behaviour and experiences—differences that could provide useful information about who is at greater potential risk of experiencing SHA (Barelds and Dijkstra 2016; Kulig et al. 2019). The observations in the present study, including the association between personality traits, such as a high degree of naivety, altruism and agreeableness, and experiencing SHA, function as input for further research into the relationship between personality and experiencing SHA in sports.

On a social level, the previous focus was mostly on the sports context as a reason for the development of SHA in sports, whereas the present study provides insight into a broader social risk profile. The finding that problems within the family context may play a role in experiencing SHA in sports is supported by previous research into risk factors for SHA in general, in which a lack of warmth from the primary caregivers and an impaired attachment were associated with a higher chance of undergoing SHA (Butler 2013; Lightfoot and Evans 2000). The relationship of trust between an athlete and their coach often replaces the poor relationship with their parents or caregivers (Cense and Brackenridge 2001). Social isolation, not only within sports, but also at school or in the neighbourhood, is also mentioned in previous research, being related to an increased chance of experiencing SHA (Finkelhor 1980). This social isolation makes the further isolation of the athlete easier for a coach (Cense and Brackenridge 2001).

The findings provide a first step towards an understanding of the biological, psychological, and social factors that could be potential risk factors for experiencing SHA in sports. In the 1990s, Cense (1997) was commissioned by NOC*NSF to conduct research into potential risk factors, which were subdivided into three areas: athlete variables (such as low self-esteem), coach variables (such as male and old age) and sports variable (such as lack of control). The results of the present study reinforce, but also extend, this previous research into risk factors, adding information about possible interactions between risk factors that lead to the susceptibility of experiencing SHA, which encourages tackling the problem at an early stage. The study by Commission de Vries (De Vries et al. 2017) focused on the policy aspects of the problem. From a macro perspective, they examined the awareness of the policy and prevention measures. Unfortunately, they did not pay attention to the development of the problem, although this is necessary for a good prevention policy.

This study had some limitations. Our study sample consisted of seven participants. In qualitative research, such a sample size is acceptable when the goal is to explore new insights into largely unknown phenomena, as is the case regarding the role of biopsychosocial factors in experiences of SHA in Dutch sports. Notwithstanding the value of our exploratory findings, we cannot be certain that with our sample—especially given the challenging recruitment for this sensitive topic—we have reached theoretical saturation (i.e., the observation that at a certain point during the data collection, subsequent interviews no longer provide new insights) (Boeije 2014). Therefore, we suggest that the presented findings need to be further examined, both in more in-depth qualitative studies and in quantitative prevalence studies among larger samples, including other relevant 'stakeholders' (e.g., parents, other family members, peers and trainers). Besides follow-up research, future research into offenders is equally important. The best preventive measure

is to prevent potential offenders from becoming offenders by the early identification of their risky behaviour.

Another limitation was the use of retrospective self-reports, so memory bias must be considered (Barusch 2011). Participants were interviewed about an event (far) in the past. Memories from the past can be limited, especially if the memory to be retrieved is decades in the past and relates to a sensitive and potentially traumatic event. It is possible that the experiences that were shared in this study were not fully complete or correct.

## 5. Conclusions

Knowledge about the factors that may represent a potential risk of experiencing SHA in sports can contribute to the development of strategies to prevent SHA and to support the victims/survivors of SHA. Of course, it should be clear that victims/survivors should never be blamed for their experience of SHA. However, the use of prevention programs fully focused on perpetrators is not a comprehensive approach to stop SHA. It is therefore important to consider biopsychosocial factors in the development of prevention programs and interventions aimed at empowering athletes and stimulating their resistance towards transgressive behaviour. Researching and drawing up a risk profile may contribute to the development of a model or interventions for early identification and protection. In practice, this requires measures to make (young) athletes more resilient against the dominant position of coaches and other authority figures within sports. Athletes should learn that it is okay to say "no" and set their own boundaries. As a result, they are also likely to develop more self-confidence and assertiveness (Cense and Brackenridge 2001). Thus, potential risk factors for experiencing SHA in sports do not appear to be limited to sports itself, but also extend to the individual and the social environment outside of sports. It is important to create awareness among sports associations, club administrators, sports coaches, parents, and athletes to recognise the circumstances which could make (young) people vulnerable to experiencing SHA in sports, in order to reduce experiences with SHA in the future. A positive, caring and safe sport climate is essential for the healthy development of young people. Only then can sports contribute to positive effects on the physical, mental and social well-being of athletes.

**Author Contributions:** Conceptualization, M.v.V., D.v.d.B. and N.S.-v.V.; methodology, M.v.V., D.v.d.B. and N.S.-v.V.; software, M.v.V.; validation, M.v.V., I.R.F., D.v.d.B. and N.S.-v.V.; formal analysis, M.v.V.; investigation, M.v.V.; resources, M.v.V., I.R.F., D.v.d.B. and N.S.-v.V.; data curation, M.v.V., and D.v.d.B.; writing original draft preparation, M.v.V. and I.R.F.; writing review and editing, M.v.V., I.R.F., D.v.d.B. and N.S.-v.V.; visualization, M.v.V.; supervision, D.v.d.B. and N.S.-v.V.; project administration, M.v.V. All authors have read and agreed to the published version of the manuscript.

**Funding:** This research received no external funding.

**Institutional Review Board Statement:** The study was conducted in accordance with the Declaration of Helsinki. As part of the regular ethical procedures, the study design and data collection procedures were evaluated against the Ethical Decision Tree and approved (Local Ethics Committee of Windesheim University of Applied Science (Zwolle, The Netherlands)).

**Informed Consent Statement:** Informed consent was obtained from all subjects involved in the study.

**Data Availability Statement:** The Dutch written report 'Het Ontstaan van Slachtofferschap Seksuele Intimidatie en Misbruik in de Nederlandse Sport: Een Triangulaire Benadering' is available on request.

**Acknowledgments:** We are grateful to all the (young) adults who participated in this study and were willing to share their experiences.

**Conflicts of Interest:** The authors declare no conflict of interest.

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
