# Peer review of "A Qualitative Exploration of a Biopsychosocial Profile for Experiencing Sexual Harassment and Abuse in Sports"

_socsci, doi:10.3390/socsci11070309_

Round 1

Reviewer 1 Report

This article makes a very important contribution by situating the issue of harassment in the field of sport. Research against sexual harassment is increasingly advancing. Sport is a field where breaking the silence is very necessary, and for this purpose listening to the voices of the victims is key. This article contributes in this sense.

The aim of exploring a biopsychosocial profile for the victimization of sexual harassment and abuse in sports is a huge goal accomplished in that regard.

The article is very well done and the qualitative methodology is very well used in this sense to meet the objectives set. The cases described in the interviews are also very valuable for this purpose.

To say something to improve, it is not very clear why the authors choose the biopsychosocial model to build on, and if this program has had any impact, social, political or scientific impact, since 2010 when it was created. It would be very interesting if the authors would discuss how these values, which they describe as biological, social and psychological, are counteracted for people who are more vulnerable to being victims of gender-based violence in sport. It would also be interesting to know if there is any difference between these values and others, or between the field of sport and other fields when it comes to sexual harassment. 

If Holland, as the authors say, has been a pioneer in raising these problems, it would be interesting to know how they are overcome; and above all, how they are prevented for people who seem to be more prone to this type of problem.

Finally, I was missing a reference on support networks, or solidarity networks of support for victimization' prevention for these victims. In the fieldwork, the people interviewed explain how, seeking for support was one of the things they did and needed. Therefore, in the sense of coping, including the variant support, and all the research on bystander intervention, will improve the paper in this sense.

Reviewer 2 Report

please see attached word file 

Round 2

Reviewer 2 Report

Much better!  Thank you for your consideration of the reviewer comments. Please see attached file for my only comment.
